# Infiltration of FoxP3+ Regulatory T Cells is a Strong and Independent Prognostic Factor in Head and Neck Squamous Cell Carcinoma

**DOI:** 10.3390/cancers11020227

**Published:** 2019-02-15

**Authors:** Imelda Seminerio, Géraldine Descamps, Sophie Dupont, Lisa de Marrez, Jean-Alexandre Laigle, Jérôme R Lechien, Nadège Kindt, Fabrice Journe, Sven Saussez

**Affiliations:** Department of Human Anatomy and Experimental Oncology, Faculty of Medicine and Pharmacy, University of Mons, 7000 Mons, Belgium; geraldine.descamps@umons.ac.be (G.D.); sophie.dupont@student.umons.ac.be (S.D.); lisa.de.marrez@gmail.com (L.d.M.); j.a.laigle@gmail.com (J.-A.L.); jerome.lechien@umons.ac.be (J.R.L.); nadege.kindt@umons.ac.be (N.K.); fabrice.journe@umons.ac.be (F.J.)

**Keywords:** HNSCC, FoxP3+ Tregs, prognosis, tumor stage, histological grade, HPV

## Abstract

Head and Neck Squamous Cell Carcinomas (HNSCC) are characterized by a large heterogeneity in terms of the location and risk factors. For a few years now, immunotherapy seems to be a promising approach in the treatment of these cancers, but a better understanding of the immune context could allow to offer a personalized treatment and thus probably increase the survival of HNSCC patients. In this context, we evaluated the infiltration of FoxP3+ Tregs on 205 human formalin-fixed paraffin-embedded HNSCC and we assessed its prognostic value compared to other potential prognostic factors, including HPV infection. First, we found a positive correlation of FoxP3+ Treg infiltration between the intra-tumoral (IT) and the stromal (ST) compartments of the tumors (*p* < 0.0001). A high infiltration of these cells in both compartments was associated with longer recurrence-free (ST, RFS, *p* = 0.007; IT, RFS, *p* = 0.019) and overall survivals (ST, OS, *p* = 0.002; ST, OS, *p* = 0.002) of HNSCC patients. Early tumor stage (OS, *p* = 0.002) and differentiated tumors (RFS, *p* = 0.022; OS, *p* = 0.043) were also associated with favorable prognoses. Multivariate analysis revealed that FoxP3+ Treg stromal infiltration, tumor stage and histological grade independently influenced patient prognosis. In conclusion, the combination of these three markers seem to be an interesting prognostic signature for HNSCC.

## 1. Introduction

Carcinomas of the upper aero-digestive tract, mostly known as head and neck squamous cell carcinomas (HNSCC), remain a major health problem in developed countries [1,2,3]. Despite the decrease of tobacco consumption and the awareness campaigns against alcohol consumption, the incidence of these cancers continues to increase steadily [4,5,6]. Some authors, especially in the United States, have proposed that this increase could be linked to the more frequent occurrence of Human Papillomavirus (HPV) infections in HNSCC, mostly in never-smoker and never-drinker young people (<45 years old) [7]. However, this is not necessarily the case in Europe, and mainly in Belgium where the incidence of HNSCC is one of the highest in the world [8]. Actually, recent studies on Belgian population showed that tobacco and alcohol remain the main risk factors for these cancers, compared to HPV infection [9]. The prognosis of HNSCC remains also very controversial in the literature. In fact, many studies suggest that HPV-related HNSCC patients have a better prognosis than non-infected patients, especially when the tumor is located in the oropharyngeal region [10,11]. However, other studies such as those conducted in our laboratory, showed that HPV infection correlates with a poorer overall survival of HNSCC patients, when looking at the other head and neck cancer regions, such as oral cavity, larynx and hypopharynx [12,13,14,15]. Therefore, it seems increasingly clear that HNSCC are characterized by a large heterogeneity in terms of tumor location and risk factors.

The immune system plays a critical role in detecting and fighting tumors. This is why immunotherapy is newly used in combination/addition with radio-/chemo-therapy to potentiate the effects of treatment and limit side effects in cancer patients [16]. In HNSCC, it is thus crucial to better understand the anti-tumor immune response and also to highlight a strong prognostic marker/signature independent of all other risk factors, such as tobacco, alcohol and HPV infection, in order to select more efficiently patients who would need additional immunotherapy modalities. Among immune cells, regulatory T lymphocytes (Tregs) maintain peripheral tolerance, preventing autoimmune reactions and chronic inflammatory diseases [17,18]. In cancers, Tregs are suppressors of anti-tumor responses, leading to tumor immune escape. The master regulator of Treg development and function is the transcription factor forkhead box P3 (FoxP3), which is used in most studies to detect Tregs population [19]. A recent meta-analysis listing 76 studies and more than 15,000 patients showed that FoxP3+ Treg infiltration in tumor is associated with patient’s survival. In cervical, renal, gastric, hepatocellular and breast cancers as well as melanomas, the authors noticed that a high recruitment of these cells is significantly associated with shorter overall and disease-free survivals. However, they obtained contrary results in head and neck, colorectal and oesophageal cancers, highlighting that a high infiltration of FoxP3+ Tregs in these cancers is associated with a longer overall survival [20]. Likewise, we recently demonstrated that a high number of FoxP3+ Tregs in tumor micro-environment is associated with longer overall and recurrence-free survivals of HNSCC patients [21].

In this study, we aimed to validate our previous results on a larger cohort of HNSCC patients. We evaluated FoxP3+ Tregs recruitment in 205 formalin-fixed paraffin-embedded (FFPE) HNSCC tumors, including HPV-negative (HPV−), HPV-positive p16-negative (HPV+ p16−) and HPV-positive p16-positive (HPV+ p16+) tumors. We evaluated FoxP3+ Treg infiltration in these three tumor subgroups and we investigated the prognostic value of this marker in intra-tumoral and stromal compartments of HNSCCs in comparison to other risk factors (e.g., tobacco, alcohol and HPV infection) and common prognostic markers (e.g., tumor stage and histological grade).

## 2. Results

### 2.1. FoxP3+ Cells Correspond to FoxP3+/CD4+ Treg Cells

First of all, we wanted to confirm that FoxP3+ cells were Treg cells by double staining 20 of our human HNSCC samples with FoxP3 and CD4 antibodies. 

Dual-immunofluorescence analysis confirmed that 94% of FoxP3+ cells (in red, nuclear staining) were also CD4+ cells (in green, membrane staining), demonstrating that our FoxP3+ population was highly composed of real Treg cells (Figure 1A).

### 2.2. FoxP3+ Treg Infiltration in Stromal and Intra-Tumoral Compartments are Significantly Correlated to Each Other

Then, we performed FoxP3 immunohistochemistry on our clinical series of 205 HNSCCs and we assessed the FoxP3+ Treg infiltration in the stromal (ST) compartment (around the tumors) and in the intra-tumoral (IT) compartment (Figure 1B). In the ST compartment, the median number of FoxP3+ Tregs was 112 (ranging from 6 to 467), while this number was 10 (ranging from 0 to 224) in the IT compartment. We previously used the median as a potential cutoff for each population (as we did in our past work) [21]. By using the median number as cutoff in this enlarged population, we obtained similarly that higher FoxP3+ Treg infiltration in the ST compartment is associated to a better prognosis, validating our first evaluation. In the current study, we used the Cutoff Finder web application [22] to calculate their optimal cutoff points and we found that 63.15 and 17.2 were the optimal cutoff values for ST and IT FoxP3+ Treg numbers, respectively. As shown by Spearman’s rho test, there is a significant positive correlation between FoxP3+ Treg infiltration in the ST compartment and in the IT compartment (correlation coefficient = 0.425, *p* < 0.0001, *N* = 170).

### 2.3. FoxP3+ Treg Infiltration Correlated with Some Clinical Characteristics

FoxP3+ Treg infiltration in both compartments was also evaluated according to patient clinical data, such as gender, tumor localization, tumor stage, tumor histological grade, tumor invasion and tumor risk factors (Table 1). We found a statistical difference between FoxP3+ Tregs in the ST compartment and gender (Mann-Whitney test, *p* = 0.044), the infiltration of these cells being higher in women than in men, but this observation was not found in the IT compartment (*p* = 0.21).

In both compartments, FoxP3+ Treg number was significantly different according to tumor localization, with the highest number in oral and oropharyngeal regions (Kruskal-Wallis test; ST, *p* = 0.031; IT, *p* = 0.024). Also, FoxP3+ Treg infiltration in the IT compartment correlated with tumor p16 status (Mann-Whitney test, *p* = 0.007) and the majority of p16+ tumors was located in the oropharyngeal region (Kruskal-Wallis test, *p* = 0.001).

### 2.4. High FoxP3+ Tregs Infiltration is Associated with Good Prognosis in HNSCC

We assessed the association between FoxP3+ Treg infiltration in ST and IT compartments with recurrence-free survival (RFS) and overall survival (OS) rate of patients with HNSCC. When studying the ST compartment, higher infiltrations of FoxP3+ Tregs (>63.15) were associated with favorable RFS (Cox regression, *p* = 0.007) (Figure 2A) and OS of patients (*p* = 0.002) (Figure 2B). In the IT compartment, a high number of FoxP3+ Tregs (>17.2) was statistically associated with a better prognosis in terms of RFS (*p* = 0.019) (Figure 2C) and OS of patients with HNSCC (*p* = 0.009) (Figure 2D).

### 2.5. Tumor Stage and Histological Grade are Both Associated with Good Prognosis in HNSCC

We also evaluated the association between patient survival and tumor stage, as well as with tumor histological grade. We did not find any significant correlation between tumor stage and RFS of patients, both when comparing stages in situ vs. I vs. II vs. III vs. IV (*p* = 0.055) and stages I/II vs. III/IV (*p* = 0.122) (Figure 3A). However, we found that patients with early stage tumors (I–II) have longer OS than patients with advanced stage tumors (III–IV) (*p* = 0.002) (Figure 3B), it was also observed when we compared stages in situ vs. I vs. II vs. III vs. IV separately (*p* = 0.001). Regarding tumor histological grade, we noticed that differentiated tumors have significantly better RFS when compared to undifferentiated tumors (*p* = 0.022) (Figure 3A), it is less significant when comparing grades undifferentiated vs. moderate vs. differentiated (*p* = 0.052). We also observed longer OS in patients with differentiated tumors when comparing to undifferentiated tumors (*p* = 0.043) (Figure 3D) or to moderate and undifferentiated ones (*p* = 0.013).

### 2.6. FoxP3+ Treg Infiltration, Tumor Stage and Histological Grade are Independent Prognostic Factor of HNSCC

Univariate and multivariate analyses were performed to assess the hazard ratio and the independent contribution of each clinical factor to RFS and OS of HNSCC patients (Table 2). Comparing various factors reported in Table 1, univariate analysis showed that FoxP3+ Treg infiltration in ST (RFS, HR = 2.1, *p* = 0.007; OS, HR = 2.2, *p* = 0.002) and IT compartments (RFS, HR = 2.0, *p* = 0.019; HR = 2.2, OS, *p* = 0.009), as well as tumor histological grade (RFS, HR = 2.0, *p* = 0.022; OS, HR = 1.8, *p* = 0.043), have a statistical impact on patient prognosis. This analysis also highlighted that tumor stage significantly influences the OS of HNSCC patients (HR = 2.2, *p* = 0.002). Of note, all of these factors have similar prognostic performance. This analysis also reported no correlation between gender, tumor invasion, tobacco/alcohol consumption, HPV infection and patient survivals (Table 2, univariate analysis).

In addition, multivariate analysis demonstrated that FoxP3+ Treg infiltration in the ST compartment (RFS, *p* = 0.05; OS, *p* = 0.015), tumor stage (RFS, *p* = 0.005; OS, *p* = 0.002) and tumor histological grade (RFS, *p* < 0.0001; OS, *p* = 0.031) are three significant prognostic factors for HNSCC that are strongly independent from each other. FoxP3+ Treg infiltration in the IT compartment also showed the same trend but became non-significant in this analysis (Table 2, multivariate analysis).

### 2.7. Combination of FoxP3+ Treg Infiltration in ST, Tumor Stage and Histological Grade Improve the Prediction of HNSCC Patient Outcome

Finally, we combined the three independent prognostic factors that were highlighted in our study in order to determine a potential prognostic signature in HNSCC (Figure 4). For each factor, a score of 1 was associated with a good prognosis meaning that the number of FoxP3+ Treg infiltration in the ST compartment is higher than 63.15, the tumor stage is at early stage I or II, and tumor histological grade reported differentiated tumors. Hence, when at least two out of three factors are of good prognosis (High score, *N* = 106), the patient survivals are significantly longer (Cox regression, *p* < 0.0001) and the patient groups are highly separated (Cox regression; RFS, HR = 5.4; OS, HR = 3.7). However, when only one or none out of the three markers (Low score, *N* = 30) are of good prognosis, HNSCC patients are significantly associated with a poor survival. Therefore, such signature bring the best prognosis information for HNSCC patients.

## 3. Discussion

Regulatory T lymphocytes (Tregs) are adaptive immune cells that contribute to tumor escape by suppressing immune anti-tumor responses. They are a minor subset of CD4+ Th lymphocytes, they constitute less than 5% of these cells in human peripheral blood. In HNSCCs, circulating and infiltrating Tregs increase during tumor development [23,24], their number being particularly high in advanced stage tumors and active disease [24,25].

In this study, we evaluated the FoxP3+ Treg infiltration in the stromal (ST) and intra-tumoral (IT) compartments of 205 HNSCC tumors. First of all, we found a positive correlation of FoxP3+ Treg recruitment between the two compartments, suggesting that Treg cells can proportionally infiltrated into the tumor. We also noticed that FoxP3+ Treg infiltration is correlated with some patient clinical characteristics. Indeed, it seems that the recruitment of these cells is correlated with gender in the ST compartment, tumor p16 status in the IT compartment and tumor localization in both compartments, which is in accordance with other studies [26,27].

Treg infiltration in tumor site has a prognostic impact that varies according to the type of cancer [20]. In HNSCCs, the prognostic impact of FoxP3+ Treg infiltration is highly controversial [28]. In fact, numerous studies underlined the poor prognosis of HNSCC patients with a high FoxP3+ Treg infiltration [29,30,31] while others, such as our laboratory, rather highlighted that a high recruitment of these cells is associated with a longer overall survival and a better tumor loco-regional control [21,23,26,27,32,33]. Indeed, we reported in a previous publication that FoxP3+ Treg number in the stromal compartment of HNSCC tumors is associated with longer overall (OS) and recurrence-free survivals (RFS) of patients [21]. We found the same results here, where FoxP3+ Tregs are associated with better RFS and OS, both in the ST and in the IT compartments, which validates that the infiltration of these cells could be a good prognostic marker for HNSCC patients. This discrepancy about the prognostic impact of FoxP3+ Treg infiltration in the literature might be explained by several factors. Indeed, studies showed that tumors located in the oropharyngeal region have a higher FoxP3+ Treg number, as we also noticed in this study, which could be explained by the richer lymphoid tissue in oropharynx than in larynx [26,27]. Moreover, depending on whether the tissue secretes IL-12 or TGF-β, FoxP3+ Tregs could be either immune suppression–competent or –incompetent, which strongly modifies the reaction of FoxP3+ Tregs to HNSCC and consequently the patient’s prognosis [34].

Our univariate Cox regression model also highlighted that tumor stage and tumor differentiation are associated with favorable prognosis in HNSCC tumors. As expected, patients with early stage HNSCC tumors (stages I–II) have longer OS than patients with advanced stage tumors (stages III–IV). The same findings were recently published in hypopharyngeal and laryngeal squamous cell carcinomas, where early stage tumors (stages I–II) presented a better overall survival than advanced stage tumors (III–IVc) [35]. Moreover, we found that tumors with differentiated histological grade have better RFS and OS than undifferentiated tumors. This is in accordance with Masoudi et al. who recently demonstrated that undifferentiated HNSCC tumors are 4 times more likely to have local recurrence and six to 15 times to develop metastases than well-differentiated tumors [36]. Likewise, our multivariate analysis confirmed our univariate analysis results, demonstrating that FoxP3+ Treg infiltration in the ST compartment, tumor stage and tumor histological grade have all a strong and independent impact on RFS and OS, showing that all could be used as prognostic markers for HNSCC patients, as published in other recent studies [37,38]. Finally, to further improve the prognostication in HNSCC patients, these three markers were combined in a signature integrating the good prognosis group of each markers (high ST Treg recruitment, early stage tumor and differentiated tumor). Indeed, when two or three out of these markers is of good prognosis, the patient outcome is significantly favorable. This emphasizes and upgrades the assumption that we previously published, namely that the combination of high FoxP3+ Treg infiltration and early tumor stage improve the prognosis of HNSCC patients [21]. We could therefore suggest that patients with a favorable prognosis have a well-differentiated early-stage tumor that is largely infiltrated with immunosuppression-incompetent FoxP3+ Tregs. Patients with such a prognostic profile could be treated effectively by immunotherapy and have a much better clinical outcome. Indeed, numerous studies showed in human laryngeal squamous cell carcinomas and in orthotopic mouse model of HNSCC that combining a Tregs inhibitor-based treatment with radiotherapy and PD-L1 blockade induces a tumor growth delay, a decrease of Treg numbers and an increased survival [39,40]. In conclusion, FoxP3+ Treg infiltration in tumor stroma, tumor stage and tumor histological grade are strong and independent prognostic markers in HNSCC. Combined together these three markers represent an efficient prognostic signature of HNSCC patients, which could be used to better manage the current treatments.

## 4. Materials and Methods

### 4.1. Patients and Clinical Data

Following our previous study [21] we obtained 95 new formalin-fixed, paraffin-embedded (FFPE) HNSCC tumors, bringing the final number of patients to 205. HNSCC tumors derived from patients who had curative surgery at CHU Saint-Pierre (Bruxelles, Belgium), Jules Bordet Institute (Bruxelles, Belgium), EpiCURA Baudour Hospital (Baudour, Belgium) and CHU Sart-Tilman (Liège, Belgium) between 2001 and 2017. This population has been classified according to HPV status (138 tumors were HPV− (73%), 26 were HPV+ p16− (14%) and 12 were HPV+ p16+ (6%)), to tobacco consumption (smokers and non-smokers) and to alcohol consumption (drinkers and non-drinkers) at the time of HNSCC diagnosis. This retrospective study has been approved by the Institutional Review Board (Jules Bordet Institute, number CE2319).

### 4.2. Determination of HPV Status

#### 4.2.1. DNA Extraction

All the FFPE samples were sectioned (5 μm of thickness), deparaffinized and digested with proteinase K overnight at 56 °C. DNA was extracted from samples by using the QIAamp DNA Mini Kit (Qiagen, Benelux, Belgium), according to the manufacturer’s recommended protocol.

#### 4.2.2. HPV Detection by Polymerase Chain Reaction (PCR) Amplification

The detection of HPV DNA was performed by PCR with GP5+/GP6+ primers (synthesized by Eurogentec, Liege, Belgium) that amplify a consensus region located within the L1 region of the HPV genome, as previously described [15].

#### 4.2.3. Real-Time Quantitative PCR Amplification of HPV Type-Specific DNA

All DNA extracts were tested for the presence of 18 different HPV genotypes using TaqMan-based real-time quantitative PCR targeting type-specific sequences of the following viral genes: 6 E6, 11 E6, 16 E7, 18 E7, 31 E6, 33 E6, 35 E6, 39 E7, 45 E7, 51 E6, 52 E7, 53 E6, 56 E7, 58 E6, 59 E7, 66 E6, 67 L1, and 68 E7 [41]. For the various real-time quantitative PCR assays, the analytical sensitivity ranged from 1 to 100 copies and was calculated using standard curves generated with plasmids containing the entire genome of the different HPV types. Real-time quantitative PCR for the detection of β-globin was performed in each PCR assay to verify the quality of DNA in the samples and to measure the amount of input DNA [42].

#### 4.2.4. p16 Immunohistochemistry

To determinate the transcriptionally activity of HPV, all samples were immunostained for p16 by using the recommended mouse monoclonal antibody (CINtec p16, clone E6H4, Ventana, Tucson, AZ, USA) and an automated immunostainer (Bond-Max, Leica Microsystems, Wetzlar, Germany). p16 expression was defined as positive when both the nucleus and cytoplasm were stained and when more than 70% of tumor cells were stained. This method has been described in our previous publication [43].

### 4.3. Evaluation of Tregs Recruitment

#### 4.3.1. FoxP3 Immunohistochemistry

Tregs were stained by using a FoxP3 mouse monoclonal antibody clone 236A/E7 (eBioscience, San Diego, CA, USA) at a dilution of 1:200. After deparaffinization with xylene and rehydratation with decreasing concentrations of ethanol, epitope retrieval was performed by immersing the samples in citrate buffer (ScyTek, Logan, UT, USA), and then by heating in micro-waves. Primary antibody was incubated during one hour at room temperature followed by a HRP-mouse secondary antibody, as recommended in the manufacturer’s protocol (CSAII kit, Dako, Glostrup, Denmark). Next, sections were reacted with phenol/fluorescyl-tyramid amplification reagent and HRP-tertiary antibody. Finally, diaminobenzidine and hydrogen peroxide were added on each tumor slide. The number of Foxp3+ cells (nuclear staining) was counted in 5 fields in each area (stromal and intra-tumoral compartments) with an Axio-Cam MRC5 optical microscope (Zeiss, Hallbergmoos, Germany) at 400× magnification. The mean of these 5 fields was then calculated and the median number of FoxP3+ cells was established for each compartment. Finally, we used the Cutoff Finder web application [22] to calculate the optimal cutoff points for stromal and intra-tumoral FoxP3+ Treg populations.

#### 4.3.2. FoxP3 CD4 Double Immunofluorescence

In order to confirm that FoxP3+ cells were Tregs, we performed a double immunofluorescence staining on our FFPE tumors by using a mix of a mouse monoclonal FoxP3 (cited above) and a rabbit polyclonal CD4 primary antibodies (Novus Biologicals, Centennial, CO, USA) both at a dilution of 1:200. As for FoxP3 immunohistochemistry, tissues were deparaffinized and rehydrated before epitope retrieval in micro-waves and incubation with FoxP3/CD4 primary antibodies. A mix of a goat polyclonal anti-rabbit Alexa Fluor^®^ 488 (Abcam, Cambridge, UK) and a goat polyclonal anti-mouse Alexa Fluor^®^ 555 (Abcam) was added at a concentration of 2 μg/mL. Finally, nuclei were stained with Vectashield DaPi (H-1000, Vector Laboratories, Peterborough, UK) and tumor slides were observed with confocal microscope Zeiss FluoView (Olympus, Berchem, Belgium).

### 4.4. Statistical Analyses

The medians of the independent data groups were compared by using nonparametric Mann-Whitney (2 groups) and Kruskal-Wallis test (>2 groups). Correlation between stromal and intra-tumoral Treg numbers was assess by Spearman’s rho test. The optimal cutoff points of the population were calculated by using the Cutoff finder web application [22]. Recurrence-free survival (RFS) and overall survival (OS) analyses were performed using Kaplan-Meier curves. Univariate and multivariate Cox regression models were applied to calculate hazard ratio, 95% confidence interval and significance, and to assess the independent contributions of each factor to the RFS and OS. *P*-values < 0.05 were considered statistically significant. All statistical analyses were performed by using the IBM SPSS Statistics 23 (IBM, Ehningen, Germany).

## 5. Conclusions

In conclusion, FoxP3+ Treg infiltration in tumor stroma, tumor stage and tumor histological grade are strong and independent prognostic markers in HNSCC. Combined together these three markers represent an efficient prognostic signature of HNSCC patients, which could be used to better manage the current treatments.

## Figures and Tables

**Figure 1 cancers-11-00227-f001:**
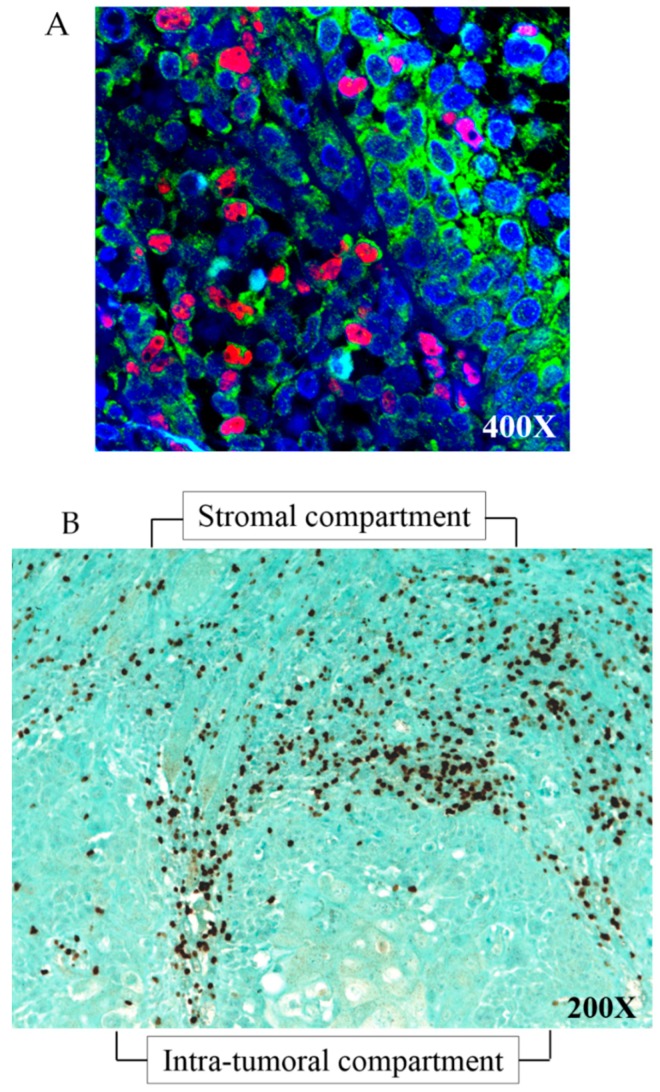
Treg immunostaining on HNSCC tumors. (**A**) FoxP3/CD4 double immunofluorescence on HNSCC tumors. Nuclear red and membrane green fluorescent markers correspond to FoxP3 and CD4 stainings, respectively; blue staining is from DAPI. (**B**) FoxP3 immunohistochemistry in stromal (ST) and intra-tumoral (IT) compartments of HNSCC tumor.

**Figure 2 cancers-11-00227-f002:**
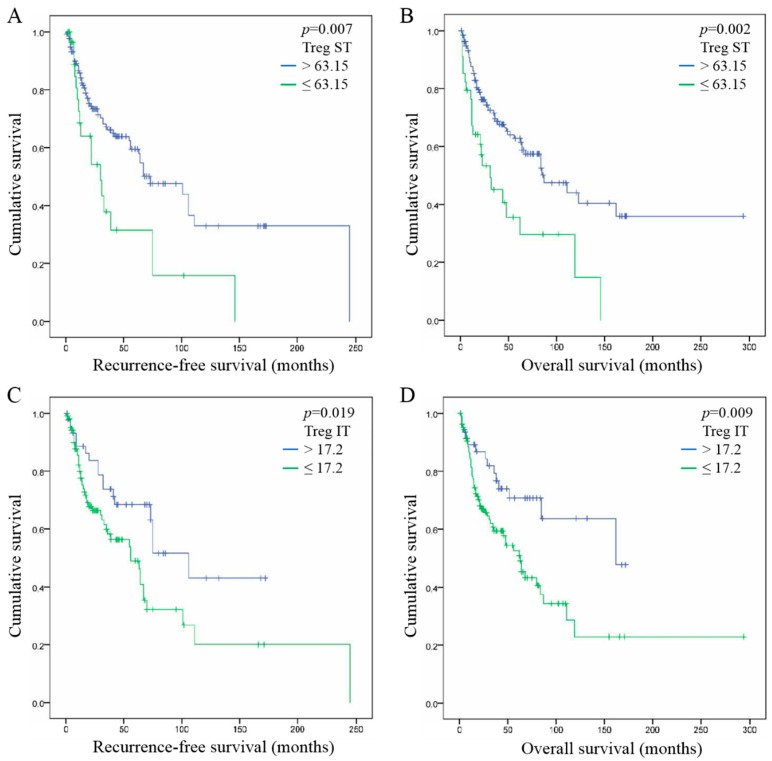
Association between FoxP3+ Treg infiltration and patient survival in HNSCC. (**A**) Association between FoxP3+ Tregs infiltration in the stromal compartment and recurrence-free survival (RFS) of patients. (**B**) Association between FoxP3+ Tregs infiltration in the stromal compartment and overall survival (OS) of patients. (**C**) Association between FoxP3+ Tregs infiltration in the intra-tumoral compartment and RFS of patients. (**D**) Association between FoxP3+ Tregs infiltration in the intra-tumoral compartment and OS of patients.

**Figure 3 cancers-11-00227-f003:**
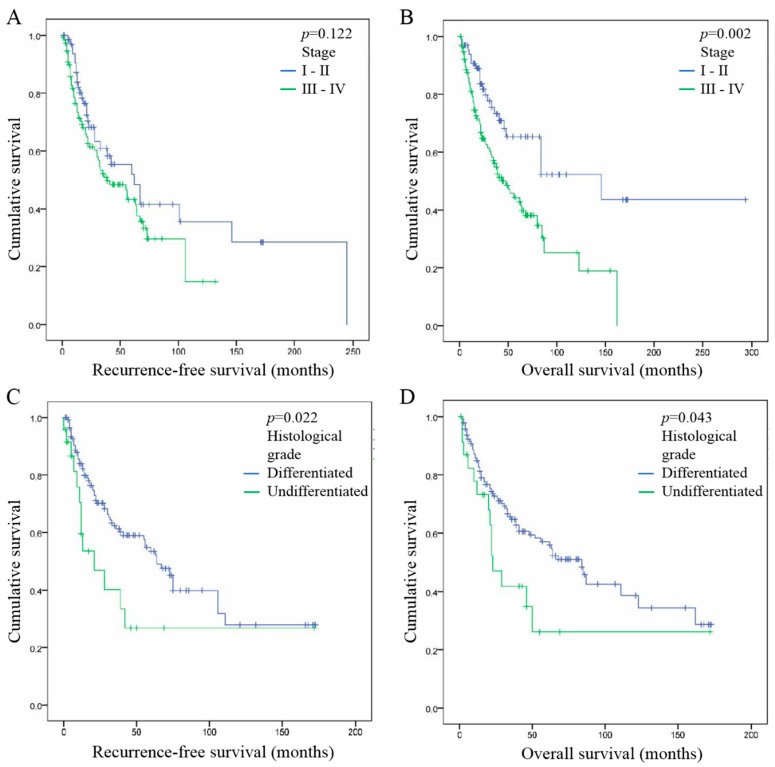
Association between tumor clinical characteristics and prognosis in HNSCC. (**A**) Association between tumor stage and recurrence-free survival (RFS) of patients. (**B**) Association between tumor stage and overall survival (OS) of patients. (**C**) Association between tumor histological grade and RFS of patients. (**D**) Association between tumor histological grade and OS of patients.

**Figure 4 cancers-11-00227-f004:**
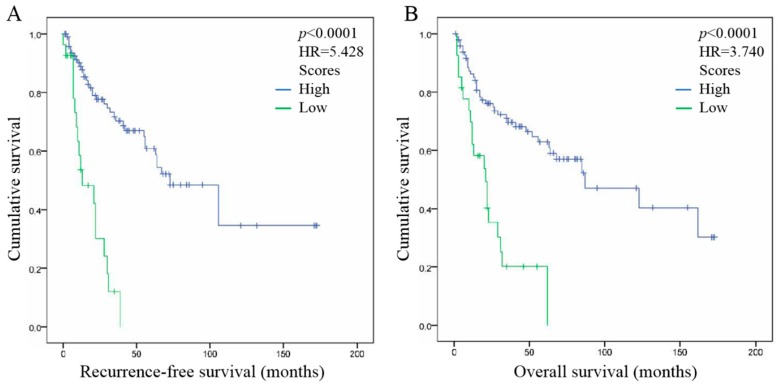
Association between combined Treg infiltration in the stromal compartment, tumor stage and tumor histological grade and prognosis in HNSCC. (**A**) Association between the three combined prognostic markers and recurrence-free survival of HNSCC patients. (**B**) Association between the three combined prognostic markers and overall survival of HNSCC patients. High score means that all the markers or two of the three markers are of good prognosis. Low score means that one of the three markers or none of the three markers are of good prognosis.

**Table 1 cancers-11-00227-t001:** Characteristics of patient population used in this study. *p* values < 0.05 are highlighted in bold.

Variables	Number of Cases (*N* = 205)	*p* Value FoxP3 in Stromal Compartment *n* > 63.15 = 38 *n* ≤ 63.15 = 164 *	*p* Value FoxP3 in Intra-Tumoral Compartment *n* > 17.2 = 119 *n* ≤ 17.2 = 52 *
Age (years)			
Median	61		
Range	29–90		
Gender		**0.044**	0.21
Male	138		
Female	67		
Anatomic site		**0.031**	**0.024**
Oral cavity	99		
Oropharynx	59		
Larynx	32		
Hypopharynx	14		
Nasopharynx	1		
Tumor stage		0.497	0.748
T1isN0	1		
T1N0	42		
T2N0	31		
T1N1	5		
T2N1	8		
T3N0	9		
T3N1	4		
T1N2	4		
T2N2	16		
T3N2	13		
T3N3	1		
T4aN0	24		
T4aN1	10		
T4aN2	19		
T4aN3	1		
T4bN2	1		
Unknown	16		
Histological grade (differentiation)		0.1	0.806
Undifferentiated	26		
Moderate	69		
Well	96		
Unknown	14		
Tumor invasion		0.238	0.649
Yes	142		
None	54		
Unknown	9		
Risk factors			
Tobacco		0.768	0.81
Smoker	160		
Non smoker	35		
Unknown	10		
Alcohol		0.732	0.72
Drinker	111		
Non drinker	73		
Unknown	21		
HPV status		0.814	0.502
Negative	127		
Positive	35		
p16 status		0.064	**0.007**
Negative	112		
Positive	14		
HPV/p16 status		0.357	0.097
Negative	127		
Positive, p16 negative	24		
Positive, p16 positive	11		
Unknown	43		
Recurrence (RFS)			
Median (months)	23		
Range (months)	1–245		
Yes	98		
None	107		
Overall survival (OS)			
Median (months)	33		
Range (months)	1–294		
Alive	106		
Dead	88		

* *N* < 205 because some tumor samples were non-evaluable for FoxP3 expression (less than 5 fields in ST and/or IT areas for counting).

**Table 2 cancers-11-00227-t002:** Correlation between patient clinical data and recurrence-free (RFS) and overall survivals (OS). *p* values < 0.05 are highlighted in bold.

*Variables*	RFS		OS	
	*HR (95% CI)*	*p* Value	*HR (95% CI)*	*p* Value
**Univariate analysis**				
Tregs ST (>63.15 vs. <63.15)	2.08 (1.22–3.57)	**0.007**	2.23 (1.35–3.69)	**0.002**
Tregs IT (>17.2 vs. <17.2)	1.96 (1.11–3.44)	**0.019**	2.19 (1.21–3.98)	**0.009**
Gender (male vs. female)	1.43 (0.95–2.15)	0.084	1.05 (0.69–1.60)	0.812
Tumor stage (I–II vs. III–IV)	1.42 (0.90–2.23)	0.122	2.20 (1.34–3.61)	**0.002**
Histological grade (differentiated vs. undifferentiated)	2.01 (1.10–3.67)	**0.022**	1.82 (1.01–3.25)	**0.043**
Tumor invasion (invasion vs. no invasion)	1.60 (0.93–2.74)	0.086	1.41 (0.85–2.33)	0.183
Tobacco (smokers vs. no smokers)	1.35 (0.77–2.36)	0.284	1.30 (0.74–2.27)	0.356
Alcohol (drinkers vs. no drinkers)	1.11 (0.72–1.72)	0.618	1.15 (0.74–1.80)	0.519
HPV status (positive vs. negative)	1.54 (0.87–2.71)	0.131	1.16 (0.69–1.95)	0.562
p16 status (positive vs. negative)	2.12 (0.76–5.91)	0.148	1.86 (0.67–5.18)	0.229
**Multivariate analysis**				
Tregs ST (>63.15 vs. <63.15)	2.24 (1.00–5.02)	**0.05**	2.32 (1.17–4.58)	**0.015**
Tregs IT (>17.2 vs. <17.2)	1.85 (0.94–3.64)	0.074	1.79 (0.92–3.47)	0.083
Tumor stage (I–II vs. III–IV)	3.33 (1.44–7.70)	**0.005**	3.72 (1.59–8.73)	**0.002**
Histological grade (differentiated vs. undifferentiated)	4.76 (2.00–11.36)	**<0.0001**	2.53 (1.08–5.88)	**0.031**

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
