# Peer review of "Infiltration of FoxP3+ Regulatory T Cells is a Strong and Independent Prognostic Factor in Head and Neck Squamous Cell Carcinoma"

_cancers, 2019, doi:10.3390/cancers11020227_

Round 1
Reviewer 1 Report
Comments:
This is a well-organized study which demonstrates FoxP3+ Regulatory T Cells are potential clinical utility in prediction to the therapy in HNSCC patients.
The paper was well written, and the data were logically presented. The conclusion was supported by the evidence collected in the work, although there are still some questions remain to be addressed:
Major comments
Include a list of number of cases and FoxP3 expression together with Table 1. This way to s easier for the reader to find this information in this article. You can provide this information in Table 1.
You have recruited n=205 HNSCC patients, but the corresponding FoxP3 expression need to be verified (using matched pairs normal mucosa). This is an important factor. Do FoxP3 expression patterns also indicate in prediction of the immunotherapy therapy in HNSCC patients?
Publicly available data of the TCGA cohort are analyzed to confirm the correlation between FoxP3 transcription and patient survival.
Is there any evidence to indicate that upregulation of FoxP3 proteins does induce the tumorigenesis of HNSCC?

Author Response
Please, see document attached

Reviewer 2 Report
In this study, the authors evaluated the infiltration of FoxP3+ Tregs on 205 human formalin-fixed paraffin-embedded HNSCC and they assessed its prognostic value compared to other 16 potential prognostic factors. The results showed that FoxP3+ Treg stromal infiltration, tumor stage and histological grade independently influenced patient prognosis and the combination of these three markers might to be a prognostic signature for HNSCC. This study designed well and the results presented clearly. However, it will be better if the author could give more words about the how the FoxP3+ Tregs is correlated with HPV status and how the HPV status is associated with prognosis in HNSCC since HPV status are getting more and more important in HNSCC.
Author Response
Please, see document attached

Reviewer 3 Report
Dear Author, Overall the mansucript is well elaborated, however it has important mistakes that must be corrected. The suggestions for improving this study are included below.
Results
The authors indicate a high positive FoxP3 population, however, in the figure 1 of the dual immunofluorescence is not adequately represented the percentage. I recommend broadly that authors include a photograph with more quality and less approaching in order to demonstrate the statement.
Page 4 line 84: “FoxP3 Treg…” The results of this section are not totally understandable according to the methodology:
1. I consider that must be indicated the site of expression by immunohistochemistry at the cell (membrane, nuclei or cytoplasm).
2. Regarding the range of ST, is not well-explained how the expression may be establish considering those ranges.
I suggest taking in account any of the following options:
a) Determine the expression by a percentage for positive cells for any compartment. If the expression is at the nuclei, performing cell counting according to the label index technique (Bologna-Molina R et al., Histopathology 2011;59(4):801-3), or
b) Indicate in the material and methods section the applied technique in the evaluation of the expression.
Figure 1B. Would it be possible that the authors include a photography with better quality? Furthermore, as possible, show the microphotographies of H&E.
I suggest taking new picture in the areas where does not exist any background.
Page 4, line 96: In the correlation tests the authors indicate a correlation coefficient of 0.425. What was the minimal range that was taken in account in this coefficient (rho=0.0-1)?
Table 1.
Please review adequately table 1 and, as possible, answer the following questions:
1. Why did not the authors make statistical analysis with age vs FoxP3?
2. Would it be possible that the authors indicate an evaluation by TNM?
3. Would it be possible that “Recurrence” and “Overall survival” be deleted and make a statistical analysis associated to OS and Recurrence vs FoxP3?
Would it be possible that the authors add the results of the section 2.4 in table 1? Otherwise, make another table with these results.
Section 2.5. I suggest the following:
1. To divide tumor stage in I,II,III and IV individually?
2. As it was told above, please divide the histological grade in four different types.
3. To associate the variables tumor stage and histological grade with FoxP3.
Discussion
Lines 184-185. I consider that these lines are repetitive in comparison with material and methods. Please review it in order to correct it or delete it.
I consider that the discussion is well redacted, and it explains adequately all the results of the study and their importance. Furthermore, it establishes an adequate comparison between other studies performed by other authors and how the results agree with this study.
Conclusion
I consider that the conclusion is not closely related with the results and discussion. I suggest making a review and establish an association between results and discussion.
Material and methods
I suggest to the authors not repeating the number of evaluated cases, due it was already included in the aim of the study.
Would it be possible that the authors indicate the approval number given by the ethics committee of Jules Bordet Institute?
As possible, please classify the HNSCC according to the histological grade: well-differentiated, moderately differentiated, poorly differentiated and undifferentiated. Otherwise, please indicate what the authors indicate with differentiated or undifferentiated.
Author Response
Please, see document attached

Round 2
Reviewer 1 Report
Authors have properly responded to the reviewer comments. Thus, I recommend that this manuscript would be valuable to be published in Cancers.
Reviewer 3 Report
Dear authors:
After making a second review of this manuscript, I consider that it has improved considerably. All the suggestions were taken in account, thus, the current study is suitable for publication.